# Performance replication of the Hospital Mental Health Risk Screen in ethnoracially diverse U.S. patients admitted through emergency care

**Eve B. Carlson** [1,2]*, **M. Rose Barlow** [1], **Patrick A. Palmieri** [3], **Lisa Shieh** [4], **Thomas A. Mellman** [5], **Erika Cooksey** [6], **Jada Parker** [7], **Mallory Williams** [6,7¤], **David A. Spain** [8]

1 Dissemination and Training Division, Department of Veterans Affairs, National Center for Posttraumatic Stress Disorder, VA Palo Alto Health Care System, Menlo Park, California, United States of America, 2 Department of Psychiatry and Behavioral Sciences, Stanford University School of Medicine, Stanford, California, United States of America, 3 Traumatic Stress Center, Summa Health, Akron, Ohio, United States of America, 4 Division of Hospital Medicine, Department of Medicine, Stanford University School of Medicine, Stanford, California, United States of America, 5 Department of Psychiatry and Behavioral Sciences, Howard University College of Medicine, Washington, DC, United States of America, 6 Center of Excellence in Trauma and Violence Prevention, Howard University College of Medicine, Washington, DC, United States of America, 7 Department of Surgery, Howard University College of Medicine, Washington, DC, United States of America, 8 Department of Surgery, Stanford University School of Medicine, Stanford, California, United States of America

¤ Current address: Department of Trauma and Burn, John H. Stroger, Jr. Hospital of Cook County, Chicago, Illinois, United States of America
* ecarlson@stanford.edu

**Data Availability Statement:** Study data and supporting files are available in the OpenICPSR repository: https://doi.org/10.3886/E208549V1

## Abstract

### Background

Patients admitted to hospitals after emergency care for injury or acute illness are at risk for later mental health problems. The American College of Surgeons Committee on Trauma Standards for care of injured patients call for mental health risk screening, and the Hospital Mental Health Risk Screen (HMHRS) accurately identified at-risk patients in a developmental study that included patients from five ethnoracial groups. Replication of these findings is essential, because initial positive results for predictive screens can fail to replicate if the items were strongly related to outcomes in the development sample but not in a new sample from the population the screen was intended for.

### Study design

Replication of the predictive performance of the 10-item HMHRS was studied prospectively in ethnoracially diverse patients admitted after emergency care for acute illness or injury in three hospitals across the U.S.

### Results

Risk screen scores and follow-up mental health outcomes were obtained for 452 of 631 patients enrolled (72%). A cut score of 10 on the HMHRS correctly identified 79% of the

**Funding:** This study was supported by the National Institute on Minority Health and Health Disparities (grant number R01MD012273 to EBC and DAS). The funders had no role in study design, data collection and analysis, decision to publish, or preparation of the manuscript.

**Competing interests:** The authors have declared that no competing interests exist.

patients who reported elevated levels of depression, anxiety, and PTSD symptoms two months post-admission (sensitivity) and 72% of the patients whose symptoms were not elevated (specificity). HMHRS scores also predicted well for patients with acute illness, for patients with injuries, and for patients who reported an Asian American/Pacific Islander, Black, Latinx, Multirace, or White identity.

## Conclusions

Predictive performance of the HMHRS was strong overall and within all five ethnoracial subgroups. Routine screening could reduce suffering and health care costs, increase health and mental health equity, and foster preventive care research and implementation. The performance of the HMHRS should be studied in other countries and in other populations of recent trauma survivors, such as survivors of disaster or mass violence.

## Introduction

Every year in the U.S., 16 to 18 million adults are admitted to hospitals after receiving emergency medical care [1–3] and are at risk for mental health problems. In injured patients, rates of depression, anxiety disorders, and/or posttraumatic stress disorder (PTSD) up to 34% have been reported [4–9]. We have recently reported comparable rates of mental health problems in patients hospitalized with acute illnesses [10]. Research in the U.S. has shown that most who experience traumatic stress never receive mental health treatment [11], and treatment delay after onset (or first diagnosis) for those who do receive care was 5 to 8 years for those with depression and 12 to 15 years for those with PTSD [11, 12]. Screening hospital patients admitted after emergency care to identify at-risk patients could reduce treatment delays and enable preventive care that could reduce or prevent mental health problems.

To address mental health risks in patients after trauma, the American College of Surgeons Committee on Trauma has set a standard for Level 1 and 2 trauma centers to screen patients for mental health risk [13]. Measures of PTSD, depression, and anxiety symptoms have been developed and validated to identify patients who have PTSD, major depression, or anxiety disorders at the time of the assessment, but these measures were not designed to predict future mental health problems and have not shown great promise as screening tools for mental health risk. Specifically, when given to hospitalized injury patients, the 20-item Posttraumatic Stress Checklist has shown mixed results to predict later PTSD symptoms, with sensitivities to predict PTSD one month later ranging from poor at .38 [14] to low at .67 [15], poor SE to predict PTSD 4 months later of .43 [14], and low to good SE to predict PTSD 6 months later of .69 [16] and .77 [15]. When given to injury patients during hospitalization, the 4-item Primary Care PTSD Screen (PC-PTSD) and the 8-item Patient Health Questionnaire both had poor sensitivity (SE = .47) to predict high PTSD symptoms six months later [17]. These findings are consistent with past research which has found that symptoms assessed in the days following exposure to traumatic stress do not accurately predict who will develop later mental health problems [18]. Therefore, our approach has been to identify which patients are *at high risk* for a mental health condition. The distinction between screens for *presence* and screens for *high risk* is common in medicine. For example, a mammogram is used to screen for *presence* of breast cancer, whereas risk assessment tools such as the Gail model are used to screen for *high risk* of breast cancer [19].

Screening tools have been created for injury patients, but performance of only a few of these have been studied in new samples beyond the sample used to develop the screen [20]. Replication of findings is essential in the development of predictive screens. Initial studies of screens can yield positive results but fail to replicate because the items were strongly related to outcomes in the development study sample but not in a new sample from the population the screen was intended for.

Given that patients hospitalized with acute illness have mental health risk comparable to that of injury patients [10], we included these patients in our study to develop the Hospital Mental Health Risk Screen (HMHRS). Similarly, given ethnoracial disparities in risk factors and in mental health outcomes after hospitalization [21, 22], in our screen development research we included sufficient numbers of patients who identified as Asian American/Pacific Islander (AAPI), Black, Latinx, Multirace, or White to inform selection of risks and items and to examine performance within the ethnoracial subgroups studied. In contrast, other screen development samples have not included patients with acute illness nor sufficient numbers of patients from diverse ethnoracial identities to inform item selection or to examine screen performance within ethnic/racial subgroups of patients [20]. Lastly, we included patients whose preferred language was Spanish or Chinese.

In the study to develop the HMHRS, data on a wide range of posttraumatic mental health risk factors were assessed in 1,320 adults admitted after emergency care for injury or acute illness in hospitals in California, Ohio, and Maryland that had Level 1 trauma centers [23]. Data from these patients who were diverse in gender, age, socioeconomic status, and ethnoracial identity was analyzed to select the most predictive risk factors, the most predictive items to assess those risk factors, and the fewest items to accurately predict later mental health. We used the total symptom burden to be predicted rather than diagnostic status because of the advantages of dimensional measurement of psychopathology and the significant limitations of traditional categorical diagnostic taxonomies [24, 25]. Specifically, PTSD, depression, and anxiety disorders are highly comorbid with about 50% of those with PTSD also meeting diagnostic criteria for depression and 20% also meeting diagnostic criteria for anxiety disorder [26]. When the goal of screening is to identify those at risk for mental health problems, a focus on diagnosis may not serve patients who have symptoms from different diagnostic categories that collectively warrant clinical attention.

In the overall sample and within ethnoracial subgroups, scores on 10 items accurately identified 75% of patients who had high levels of depression, anxiety, and PTSD symptoms two months later (sensitivity = .75) [27]. Sensitivity (SE) in ethnoracial identity groups ranged from .73 in patients who identified as White to .86 in Spanish speaking patients who identified as Latinx or Hispanic. The current study replicates the predictive performance in a new sample, which is essential to confirm that the HMHRS can reliably perform across different samples of patients.

## Methods

### Study design

This study was a prospective, longitudinal observational study designed to replicate findings on the performance of the HMHRS in a new sample of patients from three hospitals in the United States. Patients were enrolled in the study between May 12, 2021 and August 30, 2023. We recruited all patients admitted after emergency care for acute illness or injury who we could speak to before they were discharged. We recruited sufficient numbers of patients in five ethnoracial identity groups and in Spanish-speaking Latinx patients to enable performance analyses within ethnoracial subgroups and within Latinx patients who completed measures in

Spanish. We targeted all admitted patients to improve the degree to which the sample reflected the population of patients treated in U.S. hospitals in terms of age, gender, ethnoracial identity, education levels, and income levels. Mental health was assessed two months after admission because our prospective research on recovery after traumatic injury showed that, for the vast majority of those who recovered well, initial elevations in symptoms decreased to normal levels by 6 weeks [28, 29]. In addition, our pilot work that was the basis for the development of the HMHRS predicted posttraumatic symptoms at the 2-month time point [30].

## Participants

Participants were adults admitted through the emergency department in one of three hospitals (Stanford, CA, Akron, OH, or Washington, DC). Participants were between > 18 years old (mean = 50.03, SD = 16.14) and had the ability to answer spoken and/or written questions in English, Spanish, or Chinese (Traditional or Simplified). Potential study participants were identified through electronic medical records of admitted patients. Patients were included if they were treated in the emergency department, then admitted to the hospital for inpatient care. Patients were excluded if they were primarily seen for psychiatric emergency care as these patients would not be part of the population that would be screened for mental health risk. Research staff attempted to approach all eligible patients who were able to answer study questions.

## Procedures

The Stanford University Administrative Panel on Human Subjects in Medical Research approved the study and the Summa Health and Howard University IRBs determined that the study was exempt from IRB oversight. Patients were approached in their room on hospital units at least one day after admission by research assistants. At the Stanford site, bicultural and bilingual research assistants spoke English and Spanish, or English and Mandarin. At all sites, participants provided informed consent verbally after discussing the study with a research assistant and were provided with an information sheet describing the study, including study risks and benefits, payments, and the patients' right to stop participation at any time. A waiver of documentation of consent was approved by the Stanford IRB. Patients answered questions verbally, on paper, or on a tablet computer. Patients were paid $20 to answer questions at the time of enrollment and $40 for answering follow-up questions. We used follow-up methods recommended to maximize retention [31]. Participants were re-contacted after two months to answer follow-up questions over the phone, on paper, or by computer using an internet link sent to them by email.

## Measures

**During hospitalization.** Ethnoracial identity information was collected by asking patients which ethnic and racial identities applied to them from a list that included the option to specify other identities. Patients who identified as more than one ethnicity or race were categorized as Multirace.

Mental health risk was assessed with the HMHRS, a 10-item self-report measure with items assessing past mental health, expected life stress, everyday experiences of discrimination, and symptoms of depression, anxiety, PTSD [23]. Items assessing past mental health and anxiety were novel items created during the development of the HMHRS. The expected life stress item was an adapted item from the Perceived Stress Scale [32]. The discrimination item was from the Everyday Discrimination Scale [33], the depression item was from the PHQ-8 [34], the PTSD items were from the Screen for Posttraumatic Stress Symptoms (SPTSS) [35], and the

dissociation item was from the Dissociative Symptoms Scale [36]. The measure is included in the Appendix. The discrimination item was scored from 0 (never) to 5 (Almost every day). All other items were scored from 0 (None of the time) to 4 (More than half of the time). As in the development study, scores of 10 or above were used to identify those at high risk for later mental health problems.

**Follow-up.** Depression was assessed with the PHQ-8 [34, 37] and anxiety was assessed with the Generalized Anxiety Disorder-7 (GAD-7) . Symptoms of PTSD were assessed with the SPTSS, a 20-item measure based on diagnostic criteria for PTSD [35]. All three measures ask participants to report their symptoms for the past week.

Mental Health Symptoms at follow-up was defined as the sum of total scores on all items of the PHQ-8, the GAD-7, and the SPTSS. With 35 items, each scored 0–4, the range of possible scores was 0–140. To define an elevated level of Mental Health Symptoms, we identified the patients who did not report elevated depression, PTSD, or anxiety symptoms at follow-up by applying cut scores for the PHQ-8 [34] and GAD-7 [38] used in primary care and the cut score for the SPTSS based on prior research [30]. The mean score for Mental Health Symptoms for patients with no symptom elevations was 10.5 (SD = 8.8). In a normal distribution, 99.7% of data points fall within one SD of the mean, and 99.85% of data points fall at or below the value of 3 SDs above the mean. Therefore, a score of 37, which falls 3 SDs above the mean of 10.5, was estimated to represent 99.9% of the population of patients who did not have elevations in symptoms at follow-up.

## Statistical analyses

Seven participants who did not self-identify as any of the five groups were included in analyses of the entire sample but not in subgroup analyses. To include 7 patients who identified as American Indian or Alaska Native (AI/AN) in group analyses, these patients were included in a combined group (Multirace/AI/AN) based on the similarity of the AI/AN and multirace patients on levels of most risks and outcomes and the fact that 71% of the patients identifying with more than one ethnoracial identity identified as AI/AN.

Comparison of HMHRS scores across ethnoracial groups was examined using a one-way ANOVA with Tukey's HSD for post-hoc comparisons between groups. All tests were two-sided. ROC (Receiver Operating Characteristic), sensitivity, and specificity analyses were conducted to quantify performance of the HMHRS classifications to identify patients who reported elevated Mental Health Symptoms at follow-up. All statistical analyses were carried out with IBM SPSS version 29 (IBM, 2023).

## Results

We enrolled 631 patients who self-identified as one or more of ethnoracial groups (Table 1). Patients ranged in age from 18 to 87 years (mean = 50.03, SD = 16.14), and 90.4% of those who reported an identity of Latinx reported only that identity. The mean number of days between admission and study enrollment was 3.5 (SD = 3.1) with 85% of patients enrolled within 5 days of admission.

The overall follow-up rate was 72.4% (457 of 631). Analyses including follow-up data were conducted on N = 452 as there was missing data for some mental health measures at follow-up for five patients. There was no difference in retention of patients who scored as "at-risk" on the HMHRS (10 or higher; 70.3% retained) and those who scored as "low-risk" (74.0%). Similarly, there was no significant difference in HMHRS scores between those who did and did not complete the follow-up. There were differences among the five ethnoracial groups in retention,

**Table 1. Characteristics of 631 patients.**

| Characteristic | n | % |
|---|---|---|
| Gender | | |
| Male | 279 | 44.2 |
| Female | 349 | 55.3 |
| Other | 3 | 0.5 |
| Race or ethnicity | | |
| Asian or Pacific Islander (AAPI) | 52 | 8.2 |
| American Indian or Alaska Native | 1 | 0.2 |
| Black | 229 | 36.3 |
| Latinx | 115 | 18.2 |
| Multirace | 37 | 5.9 |
| White | 190 | 30.1 |
| Other | 7 | 1.1 |
| Language used | | |
| English | 567 | 89.9 |
| Spanish | 57 | 9.0 |
| Chinese (Traditional) | 2 | 0.3 |
| Chinese (Simplified) | 5 | 0.8 |
| Marital status | | |
| Single, never married | 207 | 32.8 |
| Married or living with partner | 287 | 45.5 |
| Separated or divorced | 103 | 16.3 |
| Widowed | 34 | 5.4 |
| Work status | | |
| Employed full- or part-time / self-employed | 389 | 61.6 |
| Unemployed / looking for work | 32 | 5.1 |
| Retired | 115 | 18.2 |
| Student | 9 | 1.4 |
| Other or Missing | 86 | 13.6 |
| Total yearly household income | | |
| Less than $25,000 | 253 | 40.1 |
| $25,000 - $49,999 | 142 | 22.5 |
| $50,000 - $74,999 | 74 | 11.7 |
| $75,000 - $99,000 | 42 | 6.7 |
| $100,000 or more | 113 | 17.9 |
| Missing | 7 | 1.1 |
| Education | | |
| None or elementary/primary school | 47 | 7.4 |
| High school or GED | 205 | 32.5 |
| Some college | 165 | 26.1 |
| College degree | 129 | 20.4 |
| Graduate degree | 85 | 13.5 |

$X^2$ (4) = 33.68, $p < .001$. The retention rates by ethnoracial subgroups were: 87% for AAPI, 59% for Black, 75% for Latinx, 82% for Multirace/AI/AN, and 81% for White.

HMHRS scores ranged from 0 to 41. Mean score was 10.7 (SD = 8.83), median score was 9. The percent of each ethnoracial subgroup scoring 10 or more on the HMHRS was 48% for

AAPI, 52% for Black, 41% for Latinx, 58% for Multirace, and 42% for White. However, there were no significant differences between any of the ethnoracial groups in mean score.

Table 2 shows the values in the overall sample, in injury patients, in acute illness patients, and by ethnoracial identity for Area Under the Curve (AUC), sensitivity (SE), specificity (SP), positive predictive value (PPV), and negative predictive value (NPV) and 95% confidence intervals for SE, SP, PPV, and NPV [39]. SE is the proportion of those who later had elevated Mental Health Symptoms that the screen classified as having high risk. SP is the proportion who did not later have elevated Mental Health Symptoms that the HMHRS classified as low risk. PPV is the proportion of all positive scores that were true positive scores. NPV is the proportion of all negative scores that were true negative scores.

## Discussion

In this replication study, performance of the HMHRS to identify patients who reported elevated levels of mental health symptoms at follow-up was strong, with 79% of these patients correctly identified. HMHRS scores also correctly identified 72% of the patients who did not report high levels of mental health symptoms at follow-up. Notably, sensitivity and specificity were also very good to excellent within all the ethnoracial subgroups we studied. PPVs for the ethnoracial subgroups indicate that the risk for those who screen positive is at a meaningful level, and high scores provide valuable information. NPVs for the ethnoracial subgroups were quite high, indicating that a low score on the screen is a strong indicator that the patient is at low risk for later mental health problems.

Performance overall and for most of the subgroups was superior to performance in the development study, most likely due to higher retention of at-risk patients in this study. In the prior study, 53.4% of patients scoring as "at-risk" on the screen were retained, which was

**Table 2. Performance of an HMHRS score of ten or more to predict high mental health symptoms at follow-up by ethnoracial subgroup.**

| Group | N | AUC | Sensitivity (95% CI) | Specificity (95% CI) | PPV (95% CI) | NPV (95%CI) |
|---|---|---|---|---|---|---|
| All patients | 452* | .84 | .79 (.73, .86) | .72 (.67, .77) | .58 (.52, .65) | .88 (.83, .92) |
| Injury patients | 162# | .83 | .77 (.67, .88) | .73 (.64, .82) | .64 (.53, .75) | .84 (.76, .92) |
| Acute Illness patients | 285# | .84 | .80 (.72, .89) | .71 (.65, .78) | .55 (.46, .64) | .89 (.84, .94) |
| AAPI | 45 | .84 | .83 (.62, 1.0) | .67 (.51, .83) | .48 (.26, .69) | .92 (.81, 1.0) |
| Black | 135 | .84 | .82 (.72, .92) | .70 (.60, .80) | .65 (.54, .76) | .85 (.76, .93) |
| Latinx (all) | 86 | .91 | .83 (.67, .98) | .75 (.64, .85) | .54 (.38, .71) | .92 (.85, .1.0) |
| Latinx (in Spanish) | 42 | .93 | .88 (.65, .1.0) | .76 (.62, .91) | .47 (.21, .72) | .96 (.89, 1.0) |
| Multirace/AN/AI | 31 | .94 | 1.00 (1.0, 1.0) | .70 (.50, .90) | .65 (.42, .87) | 1.00 (1.0, 1.0) |
| White | 151 | .78 | .71 (.58, .84) | .75 (.66, .83) | .57 (.44, .69) | .85 (77, .92) |

*Total N includes 2 patients who reported ethnic or racial identities that did not fit into any of the five groups reported on. #Injury or acute illness information was unavailable for 5 patients.

HMHRS, Hospital Mental Health Risk Screen; CI, confidence interval; AAPI, Asian and Pacific Islander; AI/AN, American Indian/Alaska Native; AUC, area under the curve; PPV, Positive predictive value; NPV, Negative predictive value

significantly lower than the retention rate of 68.4% for "low-risk" patients [23]. In this study, 70.3% of at-risk patients were retained, and there was no difference in retention between low-risk and high-risk patients. Similarly, HMHRS scores were significantly higher in those lost to follow-up than in those retained in the development study, but not in this study. Screen performance is better with better retention of at-risk patients because those with higher risk scores are easier to identify as they tend to have higher symptoms at follow-up.

Reasons for higher retention and better retention of at-risk patients in the replication study are important to note. In both studies, we used follow-up methods aimed at maximizing retention [31]. These included collecting follow-up data via each patient's preferred modality and repeated attempts to reach each patient for follow-up. In the replication study, we increased the number of attempts to reach each patient and had research assistants who were fluent in Spanish or Mandarin make follow-up calls to patients in their preferred language. We also used methods during recruitment to increase the chances of reaching patients for follow-up, such as giving patients a study timeline document with the follow-up date noted on it and having patients store the study phone number as a contact on their cell phone if they wished to be contacted via text.

Unfortunately, retention for Black patients in this study was lower than for other ethnoracial groups. Challenges to retention at one site included interruptions in staffing due to the pandemic and staff attrition, being unable to reach patients due to phones no longer in service, and a high proportion of patients with violent injuries (48% of injured patients). Patients with more severe injuries may have more difficulty completing follow-up measures.

Retention was higher for Black patients in this study than in the development study (59% vs 50%). This improvement may have resulted from changing one of our study sites to a historically Black college/ university (HBCU). At the new HBCU site, the staff had established connections and liaisons with community members which likely enhanced their ability to recruit and retain these patients. Lower retention of Black patients is common in medical research. One major reason for this is low levels of trust in medical researchers, which is largely attributed to past mistreatment of Black patients in medical studies, including being subjected to dangerous and unethical research procedures without their consent [40]. Other reasons for Black patients' reluctance to participate in medical research include competing demands for time, concern about stigma, and concern about discrimination by health insurance companies [41]. Research conducted in hospitals associated with HBCUs may fare better in recruiting and retaining Black patients as research participants.

There are no prior studies to compare our findings to as no screens have been developed for or tested in patients with acute illness. In addition, no prior studies have included patients that reflect the full range of ethnoracial identity diversity of the U.S. and none have reported on their performance within ethnoracial groups. Sensitivity of the HMHRS compares favorably to that of the Injured Trauma Survivors Screen, a brief screening interview conducted and scored by study staff. In a sample of 261 U.S. injury patients screened during hospitalization and assessed for mental health one to three months later, SE = .80 and SP = .66 for depression, and SE = .73 and SP = .79 for PTSD [15]. The study was limited because the sample included few patients who were not White or Black and none who were non-English speaking. In addition, the ITSS differs from the HMHRS because training is required to administer and score the ITSS, whereas the HMHRS is a self-report measure that requires no training to score.

There were a number of strengths of this study. It is the first mental health risk screen that was designed to be used with patients admitted to hospitals after emergency care for acute illness, with patients in the five largest U.S. ethnoracial subgroups, and with patients who preferred to complete a screen in Spanish or Chinese. This was also the first such screen to have its performance tested and replicated in acute illness patients, patients in the five largest U.S.

ethnoracial subgroups, and patients who preferred the Spanish language. It was also a strength of the study that we allowed patients to select their ethnoracial identity(ies) without a forced choice option for race. This approach is increasingly recommended [42] and yields results that better reflect patients' identity. This is especially true for Latinx patients as in this study 90.4% selected only a Latinx identity. It was a strength to include and test performance of the screen in patients who identify with multiple ethnoracial subgroups. Few studies include enough of these patients to inform results or test performance. Notably, in the development study, patients who reported multiple ethnoracial identities scored higher on all risks and mental health outcomes [10, 21], so these patients are among those with the highest mental health care needs. Lastly, the study recruitment methods resulted in inclusion of patients with a wide range of education and economic circumstances, which allows generalization of findings to more of the U.S. population.

In selecting the cut point score for the HMHRS, we maximized sensitivity because the "cost" of low specificity and false positive screens is low. Patients whose score is 10 or more can be told that patients who scored like they did often had later mental health symptoms of depression, anxiety, or PTSD. A graphic is available from the authors that shows scores associated with low, medium, or high risk of later mental health problems. Given the PPV and NPV findings, when patients are given screen results, the probabilistic nature of the prediction should be emphasized. It is also important to note that while the HMHRS includes the most predictive risk items we studied, these are not the only relevant risk factors, and they may not even be the most important variables influencing mental health outcomes. Some known and major risk factors, such as social support [29, 43–47], were not included because their impact cannot be accurately assessed very soon after traumatic stress.

Preventive care programs have been found to be effective for hospital injury patients [48–52] and should be offered to patients whose scores indicate they are at risk. Because the unmet need is great, preventive programs that use self-help or guided-self-help models [53] should be considered. Hospitals that do not have capacity to provide professionally-guided preventive mental health care may give patients who screen positive for risk a self-help guide for recovery [54] that also direct patients to additional self-help and professional care if needed.

Availability of an accurate and reliable predictive screen could foster research on preventive mental health care. In past research on preventive mental health care for trauma survivors, those at risk could not be identified. This meant that a large proportion of patients assigned to a control group that received no intervention recovered naturally, which may have obscured effects of interventions for those at risk. Research that included only those at risk would make it easier to detect effects of preventive care.

Limitations of this study include that we did not collect information or report on the number of patients who were recruited, but declined to participate in the study and that we were unable to collect follow-up data from all enrolled patients. No significant difference in risk scores of those retained and those lost to follow-up indicate that those retained were similar in risk to those enrolled, but the sample that performance was calculated for may have differed in unknown ways from the population of patients who would be screened. Another limitation is that the majority of Latinx patients were of Mexican heritage, and their risk screen scores may differ from those of Latinx patients who immigrated from or whose heritage is from other countries. The sample of AAPI patients was not large enough to examine performance in subgroups of AAPI patients. Studies of subgroups of Latinx and AAPI patients are needed.

## Conclusions

The performance of the HHMRS to accurately predict later mental health problems was replicated in the overall sample and in the five ethnoracial subgroups studied. The HMHRS in English (S1 File) and Spanish (S2 File) are provided in Supporting Information files, and a graphic score explainer is available from the first author (EBC). The ease of use of the HMHRS and the meaningful information it provides about mental health risk make it a valuable addition to diagnosis, treatment, and prevention efforts. Routine screening of all patients admitted after emergency care could improve health equity by ensuring that the well-being of all patients is addressed.

Availability of an accurate and reliable predictive screen could foster research on preventive mental health care and may be valuable to identify at-risk trauma survivors in other contexts. Performance of the screen could be studied in other countries and in other populations of recent trauma survivors, such as survivors of disaster or mass violence.

## Supporting information

**S1 File. Hospital Mental Health Risk Screen.**
(PDF)

**S2 File. Examen Hospitalario de Riesgos Para La Salud Mental.**
(PDF)

## Acknowledgments

We thank the patients who shared their experiences with us. We thank Rossana Castillo-Cabanas, Megan Martau, Felicia Yen, Teivon Johnson, Gabriella Amato, and Kristen Harris, who recruited patients and collected follow-up data. This work would not have been possible without their efforts.

## Author Contributions

**Conceptualization:** Eve B. Carlson, Patrick A. Palmieri, David A. Spain.

**Data curation:** Eve B. Carlson, M. Rose Barlow.

**Formal analysis:** Eve B. Carlson, M. Rose Barlow.

**Funding acquisition:** Eve B. Carlson, Patrick A. Palmieri, David A. Spain.

**Investigation:** Eve B. Carlson, M. Rose Barlow, Patrick A. Palmieri, Jada Parker, Mallory Williams.

**Methodology:** Eve B. Carlson, Patrick A. Palmieri, David A. Spain.

**Project administration:** Eve B. Carlson, M. Rose Barlow, Patrick A. Palmieri, Thomas A. Mellman, Erika Cooksey, Mallory Williams, David A. Spain.

**Resources:** Patrick A. Palmieri, Lisa Shieh, Thomas A. Mellman, David A. Spain.

**Supervision:** Eve B. Carlson, Patrick A. Palmieri, Thomas A. Mellman, Erika Cooksey, Mallory Williams.

**Validation:** Eve B. Carlson, M. Rose Barlow.

**Visualization:** Eve B. Carlson, M. Rose Barlow.

**Writing – original draft:** Eve B. Carlson, M. Rose Barlow, Erika Cooksey, Mallory Williams.

**Writing – review & editing:** Eve B. Carlson, M. Rose Barlow, Patrick A. Palmieri, Lisa Shieh, Thomas A. Mellman, Erika Cooksey, Jada Parker, Mallory Williams, David A. Spain.

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
