## [Decision Letter · Decision Letter 0]

2 Jul 2024

PONE-D-24-16440Performance replication of the Hospital Mental Health Risk Screen in ethnoracially diverse U.S. patients admitted through emergency carePLOS ONE

Dear Dr. Carlson,

Thank you for submitting your manuscript to PLOS ONE. After careful consideration, we feel that it has merit but does not fully meet PLOS ONE’s publication criteria as it currently stands. Therefore, we invite you to submit a revised version of the manuscript that addresses the points raised during the review process.

We look forward to receiving your revised manuscript.

Kind regards,

Mu-Hong Chen, M.D., Ph.D.

Academic Editor

PLOS ONE

Journal Requirements:

Reviewers' comments:

Reviewer's Responses to Questions

**Comments to the Author**

1. Is the manuscript technically sound, and do the data support the conclusions?

Reviewer #1: Yes

Reviewer #2: Yes

2. Has the statistical analysis been performed appropriately and rigorously? 

Reviewer #1: No

Reviewer #2: Yes

3. Have the authors made all data underlying the findings in their manuscript fully available?

Reviewer #1: Yes

Reviewer #2: Yes

4. Is the manuscript presented in an intelligible fashion and written in standard English?

Reviewer #1: Yes

Reviewer #2: Yes

5. Review Comments to the Author

Reviewer #1: I have some specific suggestions for improvement:

1) More information is needed regarding how participants were approached/recruited, including the proportion approached who agreed to participate.

2) Table 2 should present confidence limits for sensitivity, specificity ,PPV, and NPV.

3) Using the total score of PHQ8, GAD7, and SPTSS as an indicator of mental health need seems (at least to me unusual). Is there some evidence or rationale supporting this approach? How was the cut-point of 37 selected (and was it selected in advance)? This sum score would seem to give greater weight to the SPTSS than to the PHQ8 and GAD7 combined.

4) The discussion should address whether a new screening tool is actually needed (rather than simply relying on well-established tools such as the PHQ2, GAD2, and the PC-PTSD-5.

Reviewer #2: The current study aimed to replicate the predictive performance of the Hospital Mental Health Risk Screen (HMHRS) in a new sample of patients from three hospitals in the United States. Replication studies are crucial for validating the effectiveness and reliability of assessment tools across different patient populations. This study is straightforward and generally well-described. Methods are appropriate. I have the following comments for the authors.

1. HMHRS was created as a screening tool to identify mental health risks in patients following a traumatic event. What was the reason for having a follow-up period of two months after admission?

2. “Using this definition, an estimated 99.9% of the population of patients who did not have elevated levels of depression, anxiety, or PTSD symptoms at follow-up would be categorized as “not high” on Mental Health Symptoms.” How is the 99.9% estimation done?

3. The manifestation of PTSD can exhibit significant variation among individuals, including immediate onset, acute onset, and delayed onset. Performing HMHRS screening immediately after emergency care may be inadequate in identifying individuals who have developed acute or delayed onset PTSD.

6. PLOS authors have the option to publish the peer review history of their article (what does this mean?). If published, this will include your full peer review and any attached files.

Reviewer #1: No

Reviewer #2: No

---

## [Author Response · Author response to Decision Letter 0]

19 Aug 2024

Response to Reviewer Comments

Thank you for the opportunity to revise and resubmit this manuscript. We have addressed all of the reviewer concerns below and in the manuscript. We have uploaded the revised manuscript with and without tracked changes. We did not track formatting changes made to bring the manuscript into compliance with PLOS One style (e.g., larger font size for headers, removing degrees for authors). Page numbers below refer to pages in the manuscript that does not include tracking. 

Reviewer #1: 

1) More information is needed regarding how participants were approached/recruited, including the proportion approached who agreed to participate.

On lines 149-157, we have added details on how participants were approached/recruited, information provided to them, and payments. On lines 212-213, we reported on the timing of the screen administration (mean days) relative to admission. Unfortunately, we were not able to track patients approached who declined to participate, so we cannot provide that information. We have added this as a limitation to the study on lines 342-343. 

2) Table 2 should present confidence limits for sensitivity, specificity, PPV, and NPV.

Thank you for this suggestion. We have added the confidence intervals for SE, SP, PPV, and NPV to the table and agree that this provides valuable information about the results. We also moved the SE and SP information for injury patients and for acute illness patients into Table 2 and added AUC, PPV, NPV and confidence intervals to provide these details for those two groups. 

3) Using the total score of PHQ8, GAD7, and SPTSS as an indicator of mental health need seems (at least to me unusual). Is there some evidence or rationale supporting this approach? How was the cut-point of 37 selected (and was it selected in advance)? This sum score would seem to give greater weight to the SPTSS than to the PHQ8 and GAD7 combined.

We appreciate that using a dimensional measure is somewhat novel, and we have added an explanation on lines 107-115 in the introduction to explain the rationale for this choice. In short, we chose this approach because of the advantages of dimensional measures over diagnostic taxonomies and because responses to trauma cross diagnostic boundaries and are highly comorbid. Specifically, PTSD, depression, and anxiety disorders are highly comorbid with past research showing about 50% of those with PTSD also meeting diagnostic criteria for depression and 20% also meeting diagnostic criteria for anxiety disorder. Since the goal of screening is to identify patients with mental health risk, a focus on diagnosis seems like it would not serve patients who have symptoms from different diagnostic categories that collectively warrant clinical attention. Similarly, when the cumulative impact of symptoms is the focus, having more items that are considered symptoms of PTSD does not seem problematic.

4) The discussion should address whether a new screening tool is actually needed (rather than simply relying on well-established tools such as the PHQ2, GAD2, and the PC-PTSD-5.

On lines 78-84, we have explained that we developed a measure of mental health risk because the presence of symptoms in the days following exposure to traumatic stress do not accurately identify which patients will develop mental health problems. To further clarify, on lines 81-85, we have provided an example of screening for presence vs risk of breast cancer that is familiar in medical settings.

Reviewer #2: The current study aimed to replicate the predictive performance of the Hospital Mental Health Risk Screen (HMHRS) in a new sample of patients from three hospitals in the United States. Replication studies are crucial for validating the effectiveness and reliability of assessment tools across different patient populations. This study is straightforward and generally well-described. Methods are appropriate. I have the following comments for the authors.

1. HMHRS was created as a screening tool to identify mental health risks in patients following a traumatic event. What was the reason for having a follow-up period of two months after admission?

We have added an explanation on lines 134-138 in the methods section of the reason for assessing symptoms at two months post-admission. Our prospective research on recovery after traumatic injury showed that, for the vast majority of those who recovered well, initial elevations in symptoms decreased to normal levels by 6 weeks. In addition, our pilot work that was the basis for the development of the HMHRS predicted posttraumatic symptoms at the 2-month time point.

2. “Using this definition, an estimated 99.9% of the population of patients who did not have elevated levels of depression, anxiety, or PTSD symptoms at follow-up would be categorized as “not high” on Mental Health Symptoms.” How is the 99.9% estimation done?

We expanded the explanation for how we identified the cut score for Mental Health Symptoms on lines 185-193. We identified the patients who did not report elevated depression, PTSD, or anxiety symptoms at follow-up by applying cut scores for the PHQ-8 and GAD-7 used in primary care and the cut score for the SPTSS based on prior research. The mean score for Mental Health Symptoms for patients with no symptom elevations was 10.5 (SD = 8.8). In a normal distribution, 99.7% of data points fall within one SD of the mean, and 99.85% of data points fall at or below the value of 3 SDs above the mean. Therefore, a score of 37, which falls 3 SDs above the mean of 10.5, was estimated to represent 99.9% of the population of patients who did not have elevations in symptoms. 

3. The manifestation of PTSD can exhibit significant variation among individuals, including immediate onset, acute onset, and delayed onset. Performing HMHRS screening immediately after emergency care may be inadequate in identifying individuals who have developed acute or delayed onset PTSD.

A 2007 systematic review of research indicated that delayed onset in the absence of prior symptoms was rare, and these cases could better be understood as exacerbations or reactivation of prior symptoms. See Andrews et al. 2007: https://web.archive.org/web/20190716195953id_/https://ajp.psychiatryonline.org/doi/pdf/10.1176/appi.ajp.2007.06091491

These findings were also reflected in a change to the diagnostic criteria for PTSD from allowing a specifier of “Delayed Onset” in DSM-IV to “With delayed expression” in DSM-5. The new specifier applies when “full diagnostic criteria are not met until at least 6 months after the event (although the onset and expression of some symptoms may be immediate).”

In addition to these changes, we also updated the comparison of performance with other screens on lines 295-302 to refer to a 2021 publication (vs a 2018 publication) and noted that training is required to administer and score the ITSS. And we added an author, Mallory Williams, who worked on the study and contributed to the manuscript, but had moved and we were unable to reach him at the time of the prior submission. We since were able to get his input on the original and

---

## [Decision Letter · Decision Letter 1]

23 Aug 2024

PONE-D-24-16440R1Performance replication of the Hospital Mental Health Risk Screen in ethnoracially diverse U.S. patients admitted through emergency carePLOS ONE

Dear Dr. Carlson,

Thank you for submitting your manuscript to PLOS ONE. After careful consideration, we feel that it has merit but does not fully meet PLOS ONE’s publication criteria as it currently stands. Therefore, we invite you to submit a revised version of the manuscript that addresses the points raised during the review process. Please adequately revise your paper according to the reviewer's comments. The authors have not adequately addressed my questions and concerns:

1) The description regarding how participants were identified and recruited is not adequate. Specifically: How were potential patients identified (from admission logs, health records, etc.). What initial eligibility criteria were applied? Was every eligible patient approached - or how were patients selected to be approached? The detail here is absolutely essential, and these findings have no value or meaning without that detail.

2) The discussion does not clearly address why a new screening questionnaire is needed. It's notable that the criterion or "gold standard" in this study was a combination of screening questionnaires already in widespread use. Why would this new questionnaire be superior to or preferred over the PHQ-2/9, GAD-2/7, and the SPTSS (or PCL-5). Given the extensive data documenting the acceptability, accuracy, and utility of those measures (including data regarding performance across a wide range of people and settings), it is necessary to identify any clear advantage of this new measure.

We look forward to receiving your revised manuscript.

Kind regards,

Mu-Hong Chen, M.D., Ph.D.

Academic Editor

PLOS ONE

Reviewers' comments:

Reviewer's Responses to Questions

**Comments to the Author**

1. If the authors have adequately addressed your comments raised in a previous round of review and you feel that this manuscript is now acceptable for publication, you may indicate that here to bypass the “Comments to the Author” section, enter your conflict of interest statement in the “Confidential to Editor” section, and submit your "Accept" recommendation.

Reviewer #1: (No Response)

Reviewer #2: All comments have been addressed

2. Is the manuscript technically sound, and do the data support the conclusions?

Reviewer #1: Partly

Reviewer #2: Yes

3. Has the statistical analysis been performed appropriately and rigorously? 

Reviewer #1: Yes

Reviewer #2: Yes

4. Have the authors made all data underlying the findings in their manuscript fully available?

Reviewer #1: Yes

Reviewer #2: Yes

5. Is the manuscript presented in an intelligible fashion and written in standard English?

Reviewer #1: Yes

Reviewer #2: Yes

6. Review Comments to the Author

Reviewer #1: The authors have not adequately addressed my questions and concerns:

1) The description regarding how participants were identified and recruited is not adequate. Specifically: How were potential patients identified (from admission logs, health records, etc.). What initial eligibility criteria were applied? Was every eligible patient approached - or how were patients selected to be approached? The detail here is absolutely essential, and these findings have no value or meaning without that detail.

2) The discussion does not clearly address why a new screening questionnaire is needed. It's notable that the criterion or "gold standard" in this study was a combination of screening questionnaires already in widespread use. Why would this new questionnaire be superior to or preferred over the PHQ-2/9, GAD-2/7, and the SPTSS (or PCL-5). Given the extensive data documenting the acceptability, accuracy, and utility of those measures (including data regarding performance across a wide range of people and settings), it is necessary to identify any clear advantage of this new measure.

Reviewer #2: This study is straightforward and generally well-described. The authors responded adequately to the reviewer's comments.

7. PLOS authors have the option to publish the peer review history of their article (what does this mean?). If published, this will include your full peer review and any attached files.

Reviewer #1: No

Reviewer #2: No

---

## [Author Response · Author response to Decision Letter 1]

28 Aug 2024

Response to Reviewer Comments

Thank you for the opportunity to revise and resubmit this manuscript. We have addressed all of the reviewer concerns below and in the manuscript. Page numbers below refer to pages in the manuscript that does not include tracking. 

Reviewer comments and responses: 

1. The description regarding how participants were identified and recruited is not adequate. Specifically: How were potential patients identified (from admission logs, health records, etc.). What initial eligibility criteria were applied? Was every eligible patient approached - or how were patients selected to be approached? The detail here is absolutely essential, and these findings have no value or meaning without that detail.

Apologies for not including this detail in the prior revision. We agree that these details are important for readers to know so they can interpret the study results. On lines 153-158, we added that eligible patients were identified through electronic medical records, that patients were included who were treated in the emergency department and admitted for inpatient care, and that research staff attempted to approach all eligible patients who were able to answer study questions. An additional detail that we could include, but that seems unnecessary is that some patients were discharged before a staff member could recruit them. 

2. The discussion does not clearly address why a new screening questionnaire is needed. It's notable that the criterion or "gold standard" in this study was a combination of screening questionnaires already in widespread use. Why would this new questionnaire be superior to or preferred over the PHQ-2/9, GAD-2/7, and the SPTSS (or PCL-5). Given the extensive data documenting the acceptability, accuracy, and utility of those measures (including data regarding performance across a wide range of people and settings), it is necessary to identify any clear advantage of this new measure.

This issue was covered in more detail in our prior paper on the development of the measure, but we understand why it is important to address in the replication paper as well. We have added an explanation on lines 78-88 to clarify that measures of PTSD, depression, and anxiety have been developed and validated to assess mental health conditions at the time of the assessment, but were not designed to predict future mental health symptoms and have not shown promise of performing well to identify mental health risk. We cite specific examples of mixed and poor performance of the PCL, the PC-PTSD, and the PHQ-8 to predict later PTSD symptoms. These findings are consistent with a larger literature on Acute Stress Disorder that shows symptoms in the days after an event do not accurately predict later mental health problems. Also, Jensen et al. (2022) (citation #21 in the paper) reviewed a large number of studies of screening tools (including the PCL, PC-PTSD, and many others) and found that no brief, self-report screen screens have reliably performed well in the U.S.. In addition, no screens have been developed for or studied in acute illness patients or in U.S. ethnoracial subgroups.

---

## [Decision Letter · Decision Letter 2]

1 Sep 2024

PONE-D-24-16440R2Performance replication of the Hospital Mental Health Risk Screen in ethnoracially diverse U.S. patients admitted through emergency carePLOS ONE

Dear Dr. Carlson,

Thank you for submitting your manuscript to PLOS ONE. After careful consideration, we feel that it has merit but does not fully meet PLOS ONE’s publication criteria as it currently stands. Therefore, we invite you to submit a revised version of the manuscript that addresses the points raised during the review process.

We look forward to receiving your revised manuscript.

Kind regards,

Mu-Hong Chen, M.D., Ph.D.

Academic Editor

PLOS ONE

Journal Requirements:

Reviewers' comments:

Reviewer's Responses to Questions

**Comments to the Author**

1. If the authors have adequately addressed your comments raised in a previous round of review and you feel that this manuscript is now acceptable for publication, you may indicate that here to bypass the “Comments to the Author” section, enter your conflict of interest statement in the “Confidential to Editor” section, and submit your "Accept" recommendation.

Reviewer #1: (No Response)

2. Is the manuscript technically sound, and do the data support the conclusions?

Reviewer #1: Yes

3. Has the statistical analysis been performed appropriately and rigorously? 

Reviewer #1: Yes

4. Have the authors made all data underlying the findings in their manuscript fully available?

Reviewer #1: Yes

5. Is the manuscript presented in an intelligible fashion and written in standard English?

Reviewer #1: Yes

6. Review Comments to the Author

Reviewer #1: The claim that existing screening tools "have not shown promise of performing as brief screens for mental health risk" should either be supported by evidence or deleted.

7. PLOS authors have the option to publish the peer review history of their article (what does this mean?). If published, this will include your full peer review and any attached files.

Reviewer #1: No

---

## [Author Response · Author response to Decision Letter 2]

12 Sep 2024

Thank you for the opportunity to revise and resubmit this manuscript. We have addressed the remaining reviewer concern below and in the manuscript. 

Reviewer comments and responses: 

Reviewer #1: The claim that existing screening tools "have not shown promise of performing as brief screens for mental health risk" should either be supported by evidence or deleted.

Apologies that it was not clearer in the manuscript that the evidence to support this statement is provided in the two sentences that follow it (lines 82-88). To further clarify that the sentences that follow provide evidence to support the statement on line 81-82, we have simplified the statement and included a transition phrase “Specifically” at the start of the next sentence. We also added words in the description of supporting evidence (lines 82-88) to characterize the levels of sensitivity and specificity as poor, low, or good for readers who may not be familiar with this index of accuracy in prediction.

---

## [Decision Letter · Decision Letter 3]

17 Sep 2024

Performance replication of the Hospital Mental Health Risk Screen in ethnoracially diverse U.S. patients admitted through emergency care

PONE-D-24-16440R3

Dear Dr. Eve B. Carlson,

We’re pleased to inform you that your manuscript has been judged scientifically suitable for publication and will be formally accepted for publication once it meets all outstanding technical requirements.

Kind regards,

Mu-Hong Chen, M.D., Ph.D.

Academic Editor

PLOS ONE

Additional Editor Comments (optional):

Reviewers' comments:

Reviewer's Responses to Questions

**Comments to the Author**

1. If the authors have adequately addressed your comments raised in a previous round of review and you feel that this manuscript is now acceptable for publication, you may indicate that here to bypass the “Comments to the Author” section, enter your conflict of interest statement in the “Confidential to Editor” section, and submit your "Accept" recommendation.

Reviewer #1: All comments have been addressed

2. Is the manuscript technically sound, and do the data support the conclusions?

Reviewer #1: (No Response)

3. Has the statistical analysis been performed appropriately and rigorously? 

Reviewer #1: (No Response)

4. Have the authors made all data underlying the findings in their manuscript fully available?

Reviewer #1: (No Response)

5. Is the manuscript presented in an intelligible fashion and written in standard English?

Reviewer #1: (No Response)

6. Review Comments to the Author

Reviewer #1: (No Response)

7. PLOS authors have the option to publish the peer review history of their article (what does this mean?). If published, this will include your full peer review and any attached files.

Reviewer #1: No

---

## [Editor Report · Acceptance letter]

19 Sep 2024

PONE-D-24-16440R3 

PLOS ONE

Dear Dr. Carlson, 

I'm pleased to inform you that your manuscript has been deemed suitable for publication in PLOS ONE. Congratulations! Your manuscript is now being handed over to our production team.

Kind regards, 

on behalf of

Dr. Mu-Hong Chen 

Academic Editor

PLOS ONE